# Prognostic Value of Apparent Diffusion Coefficient (ADC) in Patients with Diffuse Gliomas

**DOI:** 10.3390/cancers16040681

**Published:** 2024-02-06

**Authors:** Marija Bušić, Zoran Rumboldt, Dora Čerina, Željko Bušić, Krešimir Dolić

**Affiliations:** 1Department of Diagnostic and Interventional Radiology, University Hospital Split, Spinčićeva 1, 21000 Split, Croatia; mbusic@kbsplit.hr (M.B.); zebusic@kbsplit.hr (Ž.B.); 2School of Medicine, University of Rijeka, Ulica Braće Branchetta 20/1, 51000 Rijeka, Croatia; zoran.rumboldt@uniri.hr; 3Department of Oncology, University Hospital Split, Spinčićeva 1, 21000 Split, Croatia; dcerina@kbsplit.hr; 4School of Medicine, University of Split, Šoltanska 1, 21000 Split, Croatia; 5University Department of Health Studies, University of Split, Ulica Ruđera Boškovića 35, 21000 Split, Croatia

**Keywords:** glioma, magnetic resonance imaging, apparent diffusion coefficient, biomarkers

## Abstract

**Simple Summary:**

This retrospective study aimed to analyze ADC values in the tissue surrounding enhancing gliomas and in the normal-appearing white matter. The goal was to find potential correlations of these values with treatment response and survival of patients. Patients, divided into short and long survival groups, underwent stereotactic biopsy or maximal surgical resection, followed by concomitant radio-chemotherapy. Baseline and follow-up MRI scans revealed significant differences in NAWM ADC values between the groups. Overall, the study suggests that ADC values in NAWM could serve as a prognostic biomarker for diffuse glioma patients.

**Abstract:**

This study aimed to evaluate potential posttreatment changes in ADC values within the tissue surrounding the enhancing lesion, particularly in areas not exhibiting MRI characteristics of involvement. Additionally, the objective was to investigate the correlations among ADC values, treatment response, and survival outcomes in individuals diagnosed with gliomas. This retrospective study included a total of 49 patients that underwent either stereotactic biopsy or maximal surgical resection. Histologically confirmed as Grade III or IV gliomas, all cases adhered to the 2016 and 2021 WHO classifications, with subsequent radio-chemotherapy administered post-surgery. Patients were divided into two groups: short and long survival groups. Baseline and follow-up MRI scans were obtained on a 1.5 T MRI scanner. Two ROI circles were positioned near the enhancing area, one ROI in the NAWM ipsilateral to the neoplasm and another symmetrically in the contralateral hemisphere on ADC maps. At follow-up there was a significant difference in both ipsilateral and contralateral NAWM between the two groups, −0.0857 (*p* = 0.004) and −0.0607 (*p* = 0.037), respectively. There was a weak negative correlation between survival and ADC values in ipsilateral and contralateral NAWM at the baseline with the correlation coefficient −0.328 (*p* = 0.02) and −0.302 (*p* = 0.04), respectively. The correlation was stronger at the follow-up. The findings indicate that ADC values in normal-appearing white matter (NAWM) may function as a prognostic biomarker in patients with diffuse gliomas.

## 1. Introduction

Gliomas are the most common primary brain neoplasms, with glioblastomas (GBM) accounting for 49.1% of malignant brain tumors [1]. GBM, the most aggressive primary brain neoplasm in the adult population, has a poor prognosis with a mean survival of 12–14 months [2,3,4]. The current standard of care includes maximum surgical resection followed by radiotherapy combined with chemotherapy (concurrent chemo-radiation) [2,4]. The scope of surgery is based on preoperative imaging and whether the tumor is located near an eloquent region of the brain [5]. The extent of resection is an important prognostic factor, as patients with a larger residual tumor volume have a shorter survival time. When surgery cannot be performed safely (because of the location or the patient’s clinical condition) a stereotactic biopsy is performed [4,5,6]. Radiotherapy allows for improved local control and increased survival [4,7]. Most patients with diffuse gliomas receive concurrent chemotherapy with an alkylating agent, primarily temozolomide, as it has a better safety profile than other alkylating agents [4,8].

Gliomas have specific growth patterns, spreading perineuronaly, perivascularly and perifascicularly [9]. Because of their invasive growth, it is essentially impossible to completely remove diffuse gliomas. Residual neoplastic cells will likely be the source of disease progression, but these areas of infiltrative growth cannot be reliably visualized on MRI [10]. Various biomarkers that might guide the treatment strategy and predict the response to therapy have been investigated; however, none were sufficiently reliable.

Brain MRI is the standard of reference for detection and evaluation of brain tumors [4,11], based on which a biopsy or a surgical resection may be performed to provide a tissue sample for the definitive pathohistological diagnosis [4]. Functional imaging methods are performed on a regular basis to better characterize neoplastic brain lesions. Diffusion MR imaging is used to evaluate the molecular function and micro-architecture of the tissues, by probing water diffusion over distances that correspond to typical cell sizes. Apparent diffusion coefficient (ADC) maps are calculated to remove the inherent T2-weighting of diffusion-weighted imaging (DWI) and represent a measure of average diffusion of water molecules within each voxel. As cell membranes are one of the factors that impede diffusion, an increase in cell density, and therefore membrane density, leads to a decrease in diffusivity. ADC maps are therefore able to differentiate highly cellular from acellular regions. Tissues with high cellularity have low ADC values (low diffusivity) as the mobility of water protons is impeded [3,12,13,14]. It has been shown that ADC values in the regions surrounding the neoplastic signal abnormality allow for differentiation of low-grade from high-grade gliomas [15]. Additionally, significantly elevated diffusivity was found in the contralateral normal-appearing white matter (NAWM) of glioma patients, grades II to IV [16].

The diffusivity of neoplastic tissue, as can be identified on MRI, has been extensively investigated and reported in the literature. We decided to analyze ADC values in the tissue surrounding enhancing gliomas in order look for potential associations with the aggressiveness of the lesions, treatment response, and survival of patients. The goal of this research was to assess whether there are posttreatment changes of ADC values in the tissue surrounding the preoperative enhancing lesion, which does not appear involved by MRI characteristics, and in the contralateral brain. The second goal was to establish whether ADC values correlate with treatment response and survival in patients with high-grade gliomas. We hypothesized that patients with more aggressive tumors and microscopic infiltrative disease would have lower ADC values in the NAWM and that these changes could serve as a potential prognostic biomarker in patients with diffuse gliomas. 

## 2. Materials and Methods

The study adhered to the principles outlined in the Helsinki Declaration, as approved by the ethical committee. 

A total of 49 patients were included in this retrospective study. All patients were diagnosed and treated from 2016 to 2021 in a single center. Patients diagnosed and treated before 2016 were excluded as the imaging studies were not available in PACS. Other exclusion factors were a lack of baseline or follow-up MRI, a lack of diffusion imaging, and prominent artifacts rendering MR images unsuitable for the measurement of ADC values. 

Patients with grade III or IV (high-grade) gliomas were selected for this study. The included patients had either a stereotactic biopsy or a maximum surgical resection following the baseline MRI. Surgical resection was performed whenever possible (tumor in a non-eloquent region and favorable clinical status of the patient). A total of 42 patients underwent surgical resection and seven patients had a stereotactic biopsy. Grade III or IV gliomas were histologically confirmed in all cases, according to the 2016 WHO classification, and corresponded to adult type diffuse gliomas by the most recent 2021 WHO classification. Following surgery, all patients received concomitant radio-chemotherapy with temozolomide and a total dose of 60 Gy. The patients underwent radiotherapy using a linear accelerator.

For the purpose of this investigation, overall survival was considered to be from the date of the initial diagnosis to the date of the last clinical follow-up. The patients were divided into two groups based on overall survival, as we had noticed a bimodal distribution with a large central gap and divided the patients accordingly into short and long survival groups. The short survival group (*n* = 39) had an overall survival ≤ 596 days and the long survival group (*n* = 10) ≥ 924 days. 

Patients’ ages ranged from 28 to 78 years. The baseline MRI was performed before any treatment and the follow-up study 1–2 months after completed treatment. 

All scans were obtained with a 1.5 T MR scanner (Magnetom Avanto; Siemens; Munich, Germany). The imaging included axial spin-echo (TR 550 ms, TE 8.7 ms, FOV 230 mm) or 3D gradient-echo (TR 1910 ms, TE 3.53 ms, FOV 256 mm) T1-weighted sequences pre- and post-contrast, axial T2-weighted sequence (TR 5000 ms, TE 96 ms, FOV 230 mm), axial FLAIR (TR 8000 ms, TE 92 ms, FOV 230 mm) and DWI/ADC maps. Echo-planar diffusion imaging was performed in the axial plane before contrast administration. Diffusion-weighted images were acquired using b values = 0 and 1000 s/mm^2^ applied in the X, Y, and Z directions. ADC maps were calculated on a voxel-by-voxel basis with the software incorporated in the MRI unit.

All scans were analyzed with the software provided by the scanner manufacturer (Syngo.via; Siemens; Munich, Germany). Each MRI study was evaluated by a radiologist. Using a freehand volume of interest tool we outlined the margins of the enhancing lesion. Two region of interest (ROI) circles were placed adjacent to the enhancing area in a random distribution (approximately 2 mm away from the enhancing margin, within the signal abnormality on FLAIR images), one ROI at least 4 cm away from the enhancement in the ipsilateral normal-appearing white matter (NAWM) and an additional ROI in the contralateral NAWM in a symmetric fashion (Figure 1). The surface area of ROI circles was 0.4–0.5 cm^2^ (Figure 1). The placement of ROIs was visually controlled between two time points to ensure consistency and to minimize potential variations. The visual control was implemented with the goal of placing the ROIs as identically as possible to the baseline MRI, to enhance the reliability and accuracy of our longitudinal assessments. The software calculated mean ADC values of the selected ROIs. 

All statistical analyses were performed using IBM SPSS Statistics software version 20 (SPSS inc., Chicago, IL, USA). The differences between groups in ADC values were determined using a t-test. A *p*-value < 0.05 was considered statistically significant. A Spearman’s rank correlation test was performed for assessment of the correlation between ADC values and survival, due to the fact that these variables do not follow a linear relationship. To compute the Spearman correlation coefficients, we assigned ranks to the ADC values and survival outcomes independently, and then calculated the correlation using these ranks. 

## 3. Results

The median overall survival was 228 days.

Mean ADC values at baseline and follow-up were higher adjacent to the enhancing lesion, as shown in Table 1.

The short survival group patients had lower mean ADC values adjacent to the enhancing lesion compared to the patients in the long survival group, both at baseline and follow-up. (Figure 2 and Figure 3). There was a statistically significant difference in one of the two measurements adjacent to the enhancing lesion between the two groups: 0.3348 (*p* = 0.013) (Table 2). There were no statistically significant differences in ADC values between ipsilateral and contralateral NAWM at baseline, −0.0404 (*p* = 0.14) and −0.0268 (*p* = 0.279), respectively. At the follow-up MRI there was a statistically significant difference in both ipsilateral and contralateral NAWM between the two groups, −0.0857 (*p* = 0.004) and −0.0607 (*p* = 0.037), respectively (Table 3).

The results showed no correlation between survival and ADC values adjacent to the enhancing lesion at the baseline or the follow-up. However, there was a weak negative correlation between survival and ADC values in ipsilateral and contralateral NAWM at the baseline with the correlation coefficient −0.328 (*p* = 0.02) and −0.302 (*p* = 0.04), respectively (Table 4). The correlation was stronger at the follow-up with the correlation coefficients −0.575 (*p* = 0) and −0.605 (*p* = 0) for ADC values in the ipsilateral and contralateral NAWM, respectively (Table 5).

## 4. Discussion

In our retrospective study, we wanted to test the hypothesis that the infiltrative growth pattern of gliomas leads to changes in the ADC values of white matter outside of the enhancing tumor mass and that these changes could serve as a prognostic biomarker for patient survival. Table 6 provides a summary of key findings from relevant studies on diffusion imaging in gliomas.

The results of our study showed that there was a statistically significant difference in ADC values adjacent to the enhancing mass between the patients stratified according to the survival times, which corresponds to the findings of Yazdani et al. [15]. Several other studies have also shown that lower diffusivity suggests a high-grade glioma, and higher ADC values are indicative of a low-grade glioma, which is in accordance with the more cellular areas exhibiting lower water mobility [13,17,18]. 

Additionally, our study found that the ADC values were lower both at baseline and follow-up in the regions immediately adjacent to the enhancing portion of the neoplasm in patients with poor survival times, which could be explained by higher cellularity and disrupted water proton mobility in these regions [13,17]. This association between tissue cellularity and ADC was evaluated by two meta analyses which provided a confirmation that ADC has an inverse correlation with cellularity in many primary tumors, including gliomas. The correlation ranged significantly in different tumors but was strong in gliomas [19,20]. Farideh et al. also tested the ADC values in the edema surrounding the neoplasm to differentiate between high-grade and low-grade gliomas and found that the values were significantly lower in high-grade glioma patients, which is similar to our findings [21]. Various molecular markers such as IDH, MGMT promoter methylation, TP53 mutation, and EGFR amplification correlate with prognosis and treatment response. It has previously been shown that IDH wild-type gliomas have a worse prognosis than IDH mutant gliomas [22]. Research by Du et al. demonstrates significantly lower ADC values of IDH wild-type gliomas in comparison with IDH mutant gliomas [23]. These findings could explain the lower ADC values found adjacent to the enhancing tumor in patients with poor survival in our research. While previous studies have shown that perilesional ADC values can differentiate high-grade gliomas from normal tissue, they were not able to distinguish neoplastic tissue from the adjacent edema [24,25]. This represents a possible limitation of diffusivity measurements in the peritumoral region, as vasogenic edema is present in many cases. In this research, two ROIs were placed adjacent to the enhancing lesion. By placing ROIs randomly, we sought to enhance the generalizability of our findings and reduce the risk of sampling bias. This was also undertaken because the infiltrative pattern of glioma growth may not follow a predictable distribution. However only one measurement showed a statistically significant difference. This could be due to the small sample size and the heterogeneity of the peritumoral tissue, which is characterized by the coexistence of vasogenic edema, marked by the accumulation of fluid in the extracellular spaces of the brain, the infiltration of malignant cells, and molecular alterations of the parenchyma [26,27]. Furthermore, as previous authors have noted, manually placed ROIs have several disadvantages such as low intra-rater reliability [28]. Our study is unique in that it sought to determine whether patients with shorter survival times had changes in ADC values that could be indicative of more aggressive neoplasms infiltrating normal brain tissue at a microscopic level. 

Horváth et al. found that ADC values of contralateral NAWM were higher in high-grade gliomas than in low-grade gliomas [16]. All of our patients had high-grade gliomas, but we also found that the patients with longer survival times had relatively lower diffusivity in both contralateral and ipsilateral NAWM, which could be due to different intrinsic neoplastic potential or less aggressive growth. The authors have suggested a breakdown of the blood-brain barrier resulting from tumor growth and possible tumor infiltration leading to those ADC changes [17,29,30]. The differences in ADC in the NAWM found in our research at follow-up are relatively small but statistically significant and could help differentiate patients with more aggressive gliomas and worse outcomes. Another study using biexponential diffusion analysis showed that the diffusion patterns in NAWM of patients with gliomas are similar to those in the clearly neoplastic tissue, suggesting infiltrative growth [31]. Studies that looked at other ways of proving diffuse glioma growth found significant decline of N-acetylaspartame (NAA) in the whole brain with MR spectroscopy in patients with newly diagnosed gliomas as well as in treated patients, which is also an indicator of diffuse tissue abnormalities that occur with glioma growth and infiltration of NAWM with malignant cells [29,32,33].

We found a weak negative correlation between survival and ADC values in NAWM both in the vicinity of the enhancing mass and on the contralateral side. This correlation was even stronger at the follow-up MRI. The observed changes in ADC values in NAWM may be due to vasogenic edema, as suggested by Horváth et al. [21]. Vasogenic edema refers to an accumulation of fluid in the extracellular spaces of the brain, occurring as a result of various pathological processes, including growth and infiltration of gliomas. The edema can alter the water proton mobility in the affected tissue and hence result in changes of ADC values, as seen in our study. The observed changes in ADC values in NAWM might suggest the presence of infiltrating glioma cells in otherwise apparently normal brain tissue on imaging studies, as suggested by Latini et al., who analyzed infiltration patterns of gliomas using electron microscopy and diffusion parameters [34]. Glioma cell invasion depends on the destruction of the extracellular matrix and the penetration of the cells between normal brain structures, leading to the accumulation of vasogenic edema and an increase in ADC values, thus more aggressive gliomas might show a greater increase in ADC values in the NAWM due to their invasion [30,35]. 

It is important to note several limitations of our study, including a relatively small sample size and lack of molecular characterization of the gliomas, as well as the fact that not all patients underwent maximum surgical resection. An increased number of measurements with a wider variety of ROI sizes and automated methods for ROI definition could also provide more accurate diffusivity information. MR spectroscopy and perfusion were not conducted for all patients, as these imaging techniques were not included in the standard MR protocol. Consequently, the limited inclusion resulted in a relatively small sample size. Further research on ADC correlation with MR perfusion and spectroscopy in the NAWM is needed as these techniques could help differentiate more aggressive tumors. Additionally, various genetic mutations and epigenetic changes can alter the patterns of glioma growth and hence affect ADC values.
cancers-16-00681-t006_Table 6Table 6Relevant ADC research, authors, main findings, and conclusions.AuthorImportant Results ConclusionsZulfiqar M, et al. (2013) [3] Inverse correlation between ADC values measured within astrocytomas and survivalLower ADC values are associated with a worse prognosis in malignant astrocytomas, independent of tumor gradeYazdani M, et al. (2018) [15]Significantly lower ADC values in the NAWM surrounding high-grade glioma than low-grade gliomaPerilesional ADC values are useful in preoperative evaluation for glioma gradeHorváth A, et al. (2015) [16]Significantly elevated ADC values in the NAWM of glioma patients compared to control subjectsHigher diffusion in normal-appearing white matter of brain tumor patients may indicate tumor infiltrationSurov A, et al. (2017) [19]Strong negative correlation between cellularity and ADC values in gliomasADC and cellularity correlation varies among different tumorsChen L, et al. (2013) [20]Strong negative correlation between the ADC and tumor cellularity, particularly in the brainMeta-analysis confirms a correlation between ADC and tumor cellularity in patientsMomeni F, et al. (2021) [21]Significantly lower ADCvalues at tumor center and edema and in patients with high-grade gliomas than those with low-grade gliomasADC values can help differentiate between low- and high-grade gliomas in the tumor itself and the surrounding edemaCatalaa I, et al. (2006) [24]Minimal ADC values in the peritumoral tissue are lower in high-grade glioma than in low-grade glioma patientsMulti-modal imaging provides valuable information for newly diagnosed cerebral gliomasCastillo M, et al. (2001) [24]Significant differences in ADC between high-grade gliomas and low-grade gliomas. Considerable overlap between ADC values in high-grade gliomas, edema and NAWMADC values helped to distinguish high-grade glioma from normal tissue but could not be used to separate high-grade glioma from surrounding edemaHorváth A, et al. (2016) [31]ADV values were significantly higher in the NAWM of glioma patients compared to controls. Globally altered diffusion parameters suggest the presence of global vasogenic edema in the NAWM of glioma patients. Alternatively, some tumor infiltration might contribute to diffusion abnormalities in the NAWM, especially in the tumor-affected hemisphere.

## 5. Conclusions

In conclusion, our study provides a valuable insight into the correlation between NAWM diffusivity and survival in patients with gliomas. Although more research is needed, our results suggest that ADC values in NAWM could serve as a prognostic biomarker in these patients. Further studies with larger sample sizes and molecular characterization of the neoplasms are needed to confirm these findings and improve our understanding of glioma growth, ultimately providing better patient care.

## Figures and Tables

**Figure 1 cancers-16-00681-f001:**
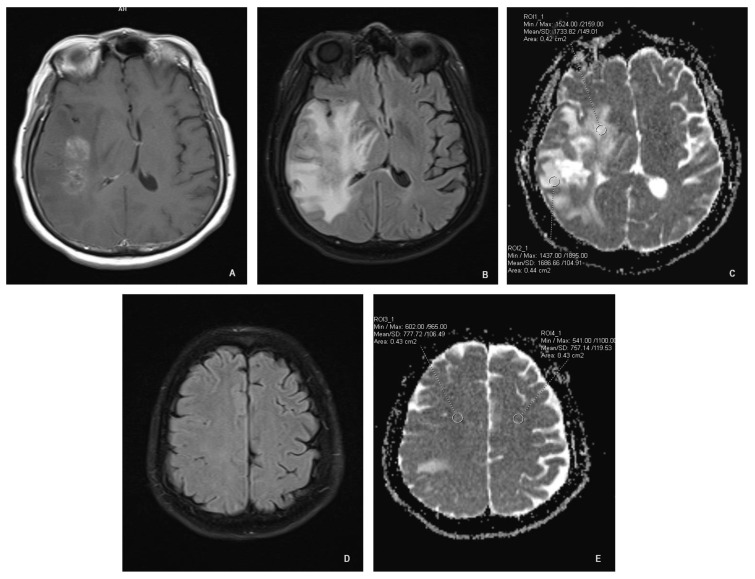
ROI placement. (**A**): T1 axial spin-echo showing the enhancing lesion; (**B**): axial FLAIR showing white matter hyperintensity adjacent to the enhancing lesion; (**C**): axial ADC map showing ROI placement adjacent to the enhancing lesion; (**D**): axial FLAIR showing normal-appearing white matter; (**E**): axial ADC at the same level as “D” showing ROI placement.

**Figure 2 cancers-16-00681-f002:**
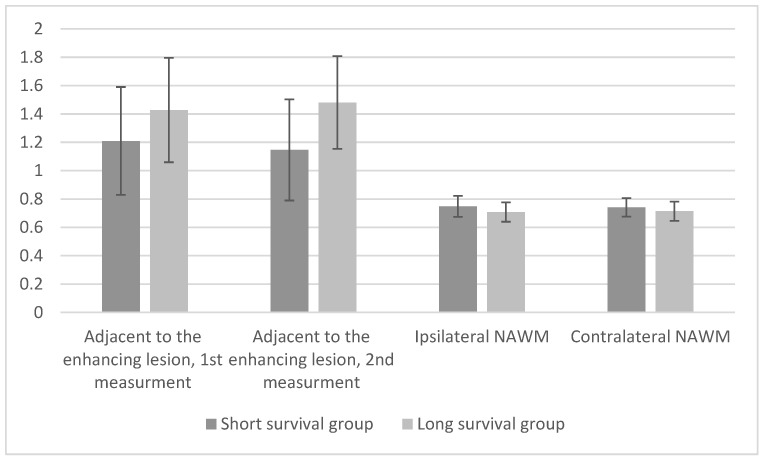
Mean ADC values for both groups at baseline.

**Figure 3 cancers-16-00681-f003:**
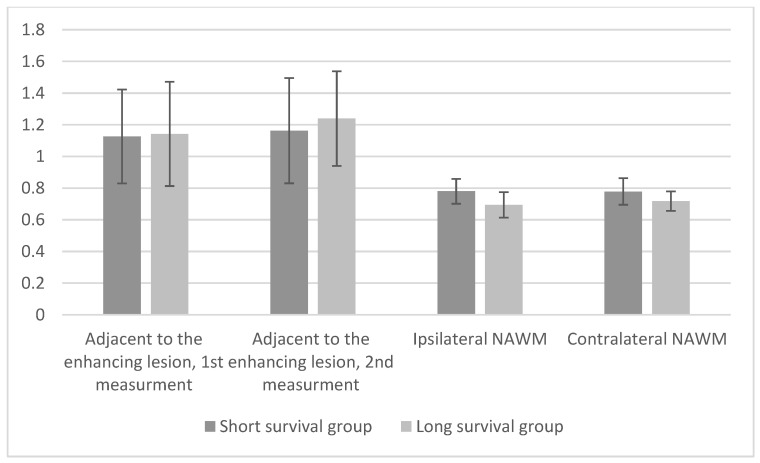
Mean ADC values for both groups at follow-up.

**Table 1 cancers-16-00681-t001:** Mean apparent diffusion coefficient (ADC) values of all patients (*n* = 49) at baseline and follow-up.

	Baseline MRI (Mean ± SD, 10^−3^ mm^2^/s)	Follow-Up MRI (Mean ± SD, 10^−3^ mm^2^/s)
Adjacent to the enhancing lesion, 1st measurement	1.2504 ± 0.3843	1.1292 ± 0.3001
Adjacent to the enhancing lesion, 2nd measurement	1.2089 ± 0.3720	1.1781 ± 0.3242
Ipsilateral NAWM	0.7407 ± 0.0738	0.7621 ± 0.0854
Contralateral NAWM	0.7358 ± 0.0663	0.7662 ± 0.0827

**Table 2 cancers-16-00681-t002:** Differences in mean ADC values between the groups at baseline.

	Short Survival Group (Mean, 10^−3^ mm^2^/s)	Long Survival Group (Mean, 10^−3^ mm^2^/s)	*p*-Value
Adjacent to the enhancing lesion, 1st measurement	1.2095	1.4726	0.1260
Adjacent to the enhancing lesion, 2nd measurement	1.1461	1.4809	0.0130
Ipsilateral NAWM	0.7482	0.7078	0.1400
Contralateral NAWM	0.7408	0.7140	0.2790

**Table 3 cancers-16-00681-t003:** Differences in mean ADC values between the groups at follow-up.

	Short Survival Group (Mean, 10^−3^ mm^2^/s)	Long Survival Group (Mean, 10^−3^ mm^2^/s)	*p*-Value
Adjacent to the enhancing lesion, 1st measurement	1.1258	1.1422	0.8800
Adjacent to the enhancing lesion, 2nd measurement	1.6252	1.2388	0.5130
Ipsilateral NAWM	0.7796	0.6938	0.0040
Contralateral NAWM	0.7786	0.7179	0.0370

**Table 4 cancers-16-00681-t004:** Correlation between overall survival and ADC values at baseline.

	Correlation Coefficient	*p*-Value
Adjacent to the enhancing lesion, 1st measurement	0.1740	0.2360
Adjacent to the enhancing lesion, 2nd measurement	0.1690	0.2500
Ipsilateral NAWM	−0.328	0.0230
Contralateral NAWM	−0.302	0.0370

**Table 5 cancers-16-00681-t005:** Correlation between overall survival and ADC values at follow-up.

	Correlation Coefficient	*p*-Value
Adjacent to the enhancing lesion, 1st measurement	−0.0030	0.9860
Adjacent to the enhancing lesion, 2nd measurement	0.0800	0.5840
Ipsilateral NAWM	−0.5750	0.0000
Contralateral NAWM	−0.6050	0.0000

## Data Availability

The data presented in this study are available on request from the corresponding author.

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
