# Peer review of "Prognostic Value of Apparent Diffusion Coefficient (ADC) in Patients with Diffuse Gliomas"

_cancers, 2024, doi:10.3390/cancers16040681_

Round 1

Reviewer 1 Report

Comments and Suggestions for Authors

Thank you for allowing me the opportunity to review your paper.  Busic et al have written a nice report regarding ADC values on MRI over time in glioma.  This study is overall straightforward and does a nice job educating the reader about the different MRI images including ADC mapping. 

I have 3 major critiques:

11)    The scope of the patient diagnoses is a little unclear: are all grades of glioma included or were high grades selected up front for the study?

22)    Nowhere is the hypothesis clearly stated.  I assume the hypothesis is that ADC values will be lower in NAWM of high grade and more aggressive gliomas representing infiltrative microscopic disease but should be specifically described in the text.  This will not be obvious to readers unfamiliar with this type of imaging.

33)    Request that more rigor be applied to the statistical analysis: 

Since the ADC values are collected in paired time points it seems a paired T test would better showcase differences than a student’s t-test. 

Error bars should be shown in the bar graphs

There is no explanation of how the correlation coefficients were calculated.

Finally, although there is a significant p value between some of the NAWM in the authors’ opinion is the change in values likely to be clinically significant.  Ie for the non-radiologist, are ADC values of 0.79 vs 0.72 expected to be very different clinically?  I think this should be broached in the discussion.

Other specific comments:

41-42   “GBM is the most ‘aggressive’” brain tumor is not adequately specific.  Many pediatric brain tumors that are not HGG are equally aggressive.  Please specify.

74        “glioma patients” please specify.  Does this mean HGG or GBM?

83        “gliomas” please specify, are all grades included?

96-97   “grade III and IV gliomas confirmed in all cases” – was this part of the selection criteria or discovered as part of the study?

100      “60Gy” is this all photon radiotherapy?  Please specify.

Figure 1 C& D For C it says 2 ROIs were placed adjacent to enhancing tissue but only one is visualized in figure.   The text in C&D is too small to read, perhaps could be included in the figure legend?  How was ROI placement controlled for between time points?  Did not see specified in methods.

133     This text appears to be attached to Figure 1, perhaps should appear before figure.

134     “groups” should be better defined.

Reviewer 2 Report

Comments and Suggestions for Authors

Manuscript:    Cancers-2848282-peer-review

Title: Prognostic Value of Apparent Diffusion Coefficient (ADC) in Patients with Diffuse Gliomas

The authors conducted a retrospective study of 49 patients with high-grade gliomas (WHO grade III and IV) and investigated possible changes in ADC values of MRI scans after treatment. Statistical comparison between patients with long and short survival time revealed significant differences between ADC values of normal-appearing white matter (NAWM) ipsilateral and contralateral, as well as a negative correlation of ADC values in NAWM with survival time. The authors hypothesize that ADC values in NAWM may act as a prognostic biomarker in patients with diffuse gliomas.

On the positive side, this study supports the important observation that NAWM is indeed of great importance for the assessment of the biological behavior of high-grade gliomas and the prognosis of patients. However, some points regarding the presentation of methods, results and discussion need to be improved before publication of the manuscript can be considered:

1) Line 121 ff: the positioning of the two regions of interest adjacent to the enhancing area needs to be explained in more detail. How do these two regions relate to each other, is one ROI more rostral, the other more caudal, or even one more medial, the other more lateral? An important finding in Table 2 is that the second measurement appears to show a significant difference between the two prognostic groups, whereas the first measurement does not. There is no discussion of this important result.

2) Figure 2: Although already mentioned in the tables, the standard deviations should be added to the bar charts. It is not usual to show only the bars for the mean values in such figures.

3) Compared to the important result of the paper, the discussion in its current form is too short and too weak. First, more references should be given to the important topic of ADC values in glioma patients, their results, and also their controversial discussion. An additional table with important results, author names, main findings and conclusions would be appropriate. Other diagnostic procedures with prognostic relevance such as MR spectroscopy, DWI, but also molecular criteria such as tumor markers, molecular tissue analyses, metabolic imaging procedures, etc. should also be included in the discussion in order to highlight the own results on ADC values within this spectrum of prognostically relevant diagnostic procedures.

Recommendation of the reviewer:

After a thorough revision of the manuscript regarding these three important points, publication of the manuscript in the journal Cancers can be reconsidered.

End of the review.
